# Blood Lead Level in a Paediatric Population of South-Eastern Spain and Associated Risk Factors

**DOI:** 10.3390/ijerph18041825

**Published:** 2021-02-13

**Authors:** Lucía Ruiz-Tudela, Maria Angeles Vázquez-López, Iciar García-Escobar, Jose Eugenio Cabrera-Sevilla, Sara Gómez-Bueno, Manuel Martín-Gonzalez, Francisco Javier Muñoz-Vico

**Affiliations:** 1Departament of Pediatric, Rafael Méndez University Hospital, 30813 Lorca, Spain; 2Departament of Pediatric, Torrecárdenas University Hospital, 04009 Almería, Spain; itxi22@hotmail.com (I.G.-E.); saragb22@hotmail.com (S.G.-B.); mmarting1959@gmail.com (M.M.-G.); 3Departament of Pediatric, Santa Lucia University Hospital, 30202 Cartegena, Spain; j_e_c_s@hotmail.com; 4Departamento of Clinic Analysis, Torrecardenas University Hospital, 04009 Almeria, Spain; fjmvla@gmail.com

**Keywords:** lead, lead poisoning, paediatrics population, sociodemografic factor, iron deficincy, erythropoyesis, Spain

## Abstract

Objective: To determine blood lead levels (BLL) in a healthy paediatric population and to analyse related sociodemographic, dietary and haematological factors. Methods: A cross-sectional study was made of 1427 healthy subjects aged 1–16 years from the city of Almería (south-eastern Spain). BLL, iron parameters and erythropoietin were determined, and sociodemographic and dietary data obtained. The study paramateters was analyses in BLL toxic and BLL no toxic group by multiple logistic regression. Results: The mean BLL was 1.98 ± 1.1 µg/dL (95% CI:1.91–2.04). For 5.7% of the population, mean BLL was 2–5 µg/dL, for 2.1% it was >5 µg/dL and for 0.15% it was >10 µg/dL. Multivariate analysis showed that immigrant origin (OR:11.9; *p* < 0.0001), low level of parental education (OR:4.6; *p* < 0.02) and low dietary iron bioavailability (OR: 3.2; *p* < 0.02) were all risk factors for toxic BLL. Subjects with toxic and non-toxic BLL presented similar iron and erythropoiesis-related parameters, except erythrocyte protoporphyrin, which was significantly higher in the BLL >5 µg/dL group. Conclusions: BLL and the prevalence of toxic BLL in healthy subjects aged 1–16 years living in south-eastern Spain are low and similar to those found in other developed countries. The factors associated with toxic BLL are immigrant origin, low level of parental education and dietary iron deficiency. The toxicity of BLL was not related to changes in the analytical parameters studied.

## 1. Introduction

Although the toxicity of lead has been known for centuries, new research suggests that even small doses can provoke alterations in various organs, especially at the neurological level [1,2]. In fact, this heavy metal does not perform any biological function in the human body, and so any dose could be considered tóxico [3,4,5].

Over the last 40 years, blood lead levels (BLL) in children have fallen dramatically, worldwide, probably due to the removal of lead from petrol, paints and other manufactured products [6], together with specific measures to limit its harmful effects [7,8]. However, population-based studies continue to report the presence of toxic levels of lead in children [4,9]. In developing or underdeveloped countries, its prevalence remains high, and so health problems persist. A recent study [10] analysed data from refugee children in the USA between 2010–2014, and recorded BLL prevalence >10 µ/dL among 16.7% of the children of Afghan origin and among 3.4% of those from Kenya. In contrast, the reported prevalence in developed countries is around 0.1% [11].

Historically, lead poisoning has been related to environmental factors, since this metal can be present in the air and soil, in the paint of old houses and in water pipes [12]. The use of traditional home remedies and cosmetics such as khol can also increase exposure to lead [12,13]. Other factors, such as poor nutritional status, iron-deficient diet, early age, pica, low socioeconomic status, living in high-risk areas, parental profession and smoking environment have also been identified [8,11,12,14].

Lead can produce alterations in iron homeostasis and interfere with erythropoiesis, although these changes have always been described for very high BLLs [15]. Other parameters recently shown to be useful in the diagnosis of iron deficiency, such as reticulocyte haemoglobin content (CHr) and serum transferrin receptor (TFr) [16], have not been tested with respect to lead poisoning in children, and very little assessment has been made of erythropoietic activity in this regard.

With regard to cut-off levels for toxicity, there is strong evidence that children’s physical and mental development can be affected even with BLL < 10 µg/dL [3,17]. In 2012, the Advisory Committee on Childhood Lead Poisoning Prevention, of the US Centers for Disease Control and Prevention (CDC), based on a population study carried out in children aged 1–5 years, established BLL >5 µg/dL, corresponding to P_97.5_ of the study sample, as the cut-off for toxicity [18]. Moreover, various population studies [19,20] have proposed lowering the toxicity limit to P_97.5_ of the values obtained in their samples, in view of the consequences observed from exposures <5 µg/dL, including intellectual dysfunction, attention disorder, hyperactivity and other behavioural problems in children. Indeed, some researchers even propose reducing the safe level to 2 µg/dL [1,2,3,21].

In recent years, studies have determined BLL in the Spanish regions of Catalonia, Madrid and the Canary Islands [22,23,24], reporting findings similar to those obtained in other developed countries [11,20]. In the southern region of Andalusia, however, no such study has yet been conducted. In view of the fact that Almería, an Andalusian province that has the third highest immigration rate in the country (persons of immigrant origin now compose 15.2% of the local population), and taking into account that ethnic origin is an important risk factor for blood lead toxicity, we consider it important to conduct a study aimed at determining BLL and the prevalence of toxic BLL among this population, and specifically, among healthy residents of Almería aged 1–16 years. In this study, we analyse relevant sociodemographic and dietary factors and assess the influence of lead on newly-proposed parameters of iron and erythropoietic activity.

## 2. Materials and Methods

This cross-sectional study was conducted of a population-based representative sample of healthy children aged 1–16 years in the city of Almería (Spain), between 2007–2009.

### 2.1. Subjects

Two different sources were used to identify the population accessible to this study: the Almería Health District and the Municipal Education Office. Data for 5453 children aged 12–47 months were obtained from the list offered by the Almería Health District, where all children living in the city are registered for metabolic screening. In addition, the Municipal Education Office provided a list of the 17,934 children aged 4–11 years and the 9823 adolescents aged 12–16 years then attending public and private primary and secondary schools. Assuming a BLL prevalence of 2%, a 95% confidence interval and 1% precision, we calculated that 895 children would be required for the study group. The children in question were selected by multistage probability sampling. For those aged 12–47 months, four of the nine health care centres were randomly selected, from which participants were recruited according to population density. For the children aged 4–16 years, six primary and six secondary schools, and two classes per grade and school, were selected. All students in these classes were invited to participate. Of the 1767 children invited to take part in the study, 17.4% refused and the remainder (1459 children) were enrolled.

All participants were resident in the city of Almería and were aged 1–16 years. Subjects with systemic or haematological disease relevant to the study parameters (according to the information provided by the children’s parents or legal guardians) were excluded.

Signed informed consent to participate was obtained from the parents or legal guardians and also from the participants who were aged 12 years or older when data collection began. All aspects of this study complied with the stipulations of the 1975 Helsinki Declaration and it was approved by the Research and Ethics Committee of Torrecárdenas Hospital (Almería).

### 2.2. Data Collection

In every case, a complete physical examination was performed. Anthropometric measurements (weight and height) were obtained with the subject barefoot and wearing light clothing. For children ≥2 years old, a Soehnle Professional 2755 digital balance (accuracy ±100 g) and an Asimed stadiometer (accuracy ±1 mm) were used, and for infants aged <2 years, data were obtained with a Seca 345 digital baby scale (accuracy ±10 g) and a Seca 207 baby measuring rod (accuracy ±1 mm). The body mass index (BMI) was calculated as weight (kg)/height^2^ (m). Overweight and obesity in children ≥2 years were defined according to the criteria proposed by the International Obesity Task Force [25], while for those aged <2 years, the equivalent BMI percentile values according to the 2010 Spanish Growth Study were used [26].

The children’s parents were asked to complete a questionnaire about their sociodemographic variables, including their origin (Spanish or immigrant), parental education level and their children’s dietary habits. Socioeconomic level was categorised as Class1 (High-income), Class 2 (Medium-income) or Class 3 (Low-income), according to employment status, adapted from the European Socioeconomic Classification [27]. Parental education level was classified into two categories: no formal education or only primary education vs. secondary or university education. Information about the children’s dietary habits was obtained using a semi-quantitative food frequency questionnaire based on the Andalusian Public Health Survey (*Encuesta Andaluza de Salud Pública*) for children and adolescents [28]. The dietary survey identifies the consumption of fruit, vegetables, meat, fish, milk and eggs. Low iron bioavailability diet was defined as the consumption of meat, fish, fruit and vegetables less than twice weekly [29].

The interviewers and researchers were all physicians who had previously completed an appropriate training and methodological standardisation programme.

### 2.3. Data Measurement

Venous blood samples were collected from fasting children. The following parameters were determined: haemoglobin, red cell indices and reticulocyte haemoglobin content (CHr) using an ADVIA-120 counter (Siemens Healthcare Diagnostics, New York, NY, USA); erythrocyte protoporphyrin (EP) by fluorometric assay [30]; serum ferritin by the immunoturbidimetric method, using Tina-Quant-Ferritin kits purchased from Roche Diagnostics GmbH (Boehringer Mannheim); serum transferrin receptor (sTfR) by immunoturbidimetric assay using the Quantex sTfR kit (Biokit SA, Barcelona, Spain). The sTfR-F Index was calculated as sTfR/log ferritin [31].

BLL was determined by graphite furnace atomic absorption spectrometry, using a Perkin Elmer model AA800 analysis instrument with Zeeman background correction. In this process, a linear calibration line is constructed with standards of 25 and 50 ppb, prepared with a solution of ammonium dihydrogen phosphate and magnesium nitrate, diluted with 0.2% nitric acid. This solution is used as a matrix modifier and as the solvent for blood samples and internal controls. A small amount of the sample is automatically introduced into the graphite chamber, where it is dried, mineralised and finally atomised in a suitable cycle of temperatures. The results of this process are obtained in absolute numerical values, with a limit of 1 µg/dL. In our analysis, values ≥ 5 µg/dL are considered toxic and those of 2–5 µg/dL are considered subclinical intoxication.

Iron deficiency is defined as any of the following: (a) iron store depletion (low ferritin); (b) iron-deficient erythropoiesis (high sTfR-F index or ≥2 parameters among low ferritin, low CHr and high sTfR); (c) iron deficiency anaemia, defined as iron deficient erythropoiesis and low haemoglobin, with cut-offs established for each age group according to the criteria of Vazquez et al. [16].

### 2.4. Statistical Analysis

Data normality was evaluated using the Kolmogorov-Smirnov test. Non-normally distributed variables were log transformed to achieve normality before analysis. The quantitative variables are expressed as mean ± standard deviation (SD), stating 95%CI and the qualitative variables as percentages. Student’s *t* test was applied to compare the mean values for BLL, and analysis of variance (ANOVA) was used to compare >2 groups. When results were significant, differences between groups were identified using the Bonferroni post-hoc test. The prevalence of toxic BLL (>5 µg/L) and of subclinical intoxication (>2–5 µg/dL) was also calculated. The odds ratio (OR) and the corresponding 95% CI were used to determine the association between toxic BLL, the individual and dietary characteristics of the subjects and the socioeconomic and cultural characteristics of the parents. Finally, multivariate logistic regression analysis was performed to identify factors independently associated with toxic BLL. Statistical significance was defined as *p* < 0.05 for all cases.

## 3. Results

The study sample consisted of 1459 infants and children aged 1–16 years. Of these, 32 were subsequently excluded due to the missing data. Finally, therefore, 1427 subjects were included in the analysis. The mean age of this population was 8.3 ± 4.5 years (95% CI: 8.0–8.5 years). Table 1 shows the characteristics of those included and the mean values of the analytical parameters determined. The percentage of immigrant subjects was 10.7%, somewhat lower than expected.

The mean BLL were 1.98 ± 1.1 µg/dL (95% CI: 1.91–2.04), with a reference interval (P_2.5_–P_97.5_) of 1–4 µg/dL. Thirty-one subjects (2.1%) presented BLL ≥ 5 µg/dL, 82 (5.7%) presented values between 2–5 µg/dL and only 2 (0.15%) presented BLL ≥10 µg/dL.

The anamnesis and physical examination revealed no symptoms or signs related to lead toxicity. 14.8% of the subjects presented pica (the habit of eating non-food items). None of them presented BLL >5 µg/dL. Table 2 shows the mean BLL obtained and the prevalence of toxic BLL, according to the different variables analysed. No significant differences were observed in relation to gender, nutritional status or presence of iron deficiency. The children aged 12–16 years presented significantly higher BLL, as did those of immigrant origin, those of low socioeconomic level, those whose parents had a low level of education and those whose diet was characterised as providing low iron bioavailability. All of these variables were identified as risk factors for presenting BLL ≥5 µg/d.

Regarding the analytical parameters studied (Table 3), the mean values obtained were similar among subjects with normal and toxic BLL, except for erythrocyte protoporphyrin, which was significantly higher in the toxic BLL group.

In the multiple logistic regression analysis, age and socio-economic level lost statistical significance (Table 4).

## 4. Discussion

Despite the primary prevention measures taken to reduce environmental emissions, the danger of lead poisoning remains present.

Our main study aim was to determine BLL and the prevalence of toxic BLL in healthy children. The study group was drawn from Almería, a city in south-eastern Spain with a wide representation of social classes and ethnic groups, where immigration has increased considerably in recent years.

In Spain, levels of lead exposure have fallen significantly [23]. An earlier study [32] reported mean values of 22 µg/dL in 1242 adults and children, of whom 23.5% presented values >25 µg/dL. More recently, however, Llopis et al. [33] studied prenatal lead exposure and recorded a mean value in cord blood of 1.1 µg/dL. Only 6% of these neonates presented values >2 µg/dL. Bas et al. [24] obtained undetectable BLL in 74% of the study population and only in 4% were >5 µg/dL. The mean BLL obtained in our study was 1.98 ± 1.91 µg/dL (P_97.5_: 4 µg/dL), and 2.1% of the subjects presented toxic BLL (>5 µg/dL), values similar to those recorded in paediatric populations in other developed countries [11,20,34]. However, we consider it worrying that 5.7% of the subjects presented BLL between 2 and 5 µg/dL, as many studies have observed neurological alterations in children with values <2 µg/dL [1,3,35,36]. Furthermore, renal, cardiovascular and reproductive alterations have been found with BLL <5 mcg/dL [17,37], which highlights the interest of our own investigation. In the present study, neither the parents’ responses to the questionnaire nor the physical examination revealed symptoms suggestive of lead poisoning in any of the subjects analysed.

Various demographic, socioeconomic, cultural and dietary factors seem to influence the BLL obtained and the risk of presenting toxic BLL. With respect to age, studies have shown that younger children are at greater risk of presenting elevated BLL [9,10,13], especially infants aged <2 years, probably related to the infants’ close contact with the environment, hand-to-mouth contact and predisposition to pica [4,14]. However, in this respect, Charalambous et al. [38] found no significant differences. Our own bivariate analysis showed that the group aged 12–16 years had the highest BLL and the highest prevalence of toxic BLL (Table 2). However, this association disappeared in the multivariate analysis, probably because this older group contained the highest proportion of immigrant population, who presented significantly higher BLL than the population of Spanish origin and were also at significantly greater risk of presenting toxic BLL. The association between BLL and immigrant origin persisted in the multivariate study (Table 4). Of the 52 Spanish provinces, Almería has the third highest percentage of immigrant population (15.20%). Of the latter, almost 30% are of Moroccan origin, 11.63% are Romanian and 9.45% are Ecuadorian. All three countries have high BLL [8], and research has shown that residence in a country where the risk of lead poisoning is known to be high can increase the risk of toxic BLL by eleven times [5]. Furthermore, this increased risk may be perpetuated if immigrants from these countries consume products imported from their country of origin. In countries where the risk of lead poisoning is greatest, there is often little control in food manufacturing [5]. In this respect, canned products, spices, cosmetics and even medicines or home-fashioned remedies based on uncontrolled production are strongly associated with the risk of lead poisoning [12]. The physical characteristics of the family home may also represent a significant risk. Thus, immigrants tend to live in marginal neighbourhoods, where buildings are old and/or poor quality, where the paintwork may be deteriorated and the water supply channelled via lead pipes [5,12]. Finally, there is strong evidence that lead is transferred to the foetus via the placenta [21,39], thus creating a transgenerational problem. Prenatal exposure is known to influence neurodevelopment. Some studies have found lower IQs among patients exposed to lead [2,21], while others conclude that neurocognitive impairment is greater with early postnatal exposure [40].

In our study, the subjects with low socioeconomic status recorded significantly higher BLL and a higher risk of toxic BLL. Previous studies, too, have associated low socioeconomic status with jobs presenting an increased risk of lead exposure [41]. Moreover, undocumented immigrants have more difficulty choosing work, and this factor can push them towards occupations where the exposure to lead may be greater [12], working as mechanics, possibly handling lead batteries, or in contact with lead-contaminated soil or materials [38]. The lead that is handled at work may subsequently be transported into the worker’s home, exposing family members to the same risk [9]. However, despite the evidence of prior research, in our case the multivariate analysis did not confirm any association between socioeconomic status and toxic BLL, possibly due to the small number of subjects comprising the toxic BLL group.

The parents’ level of education is another relevant factor. In our study, the subjects whose parents had few educational qualifications were at greater risk of toxic BLL. This finding corroborates Braun et al. [42], who observed an association between the BLL and low educational level of the mother, probably due to her lower ability to implement measures within the household to reduce lead exposure. Another study of this question, however, found no significant differences in this respect [9].

Currently in our country there are no specific health programmes for the determination of lead levels in children. In other countries, such as the United States, there are specific programmes, with the implementation of educational and preventive measures [8]. Taking into account the high percentage of immigration in our country, we suggest that blood lead testing should be carried out as part of the care of immigrant children, as well as instilling measures to reduce lead exposure.

An iron-deficient diet is known to favour lead poisoning in children [43,44], and an inverse relationship between BLL and dietary iron has been described [9,43]. In our study, subjects whose diets were characterised by low iron bioavailability had higher BLL and an increased risk of toxic BLL. Lead is known to influence iron homeostasis at different levels. Both metals are absorbed by the same transporter in the intestine. Diets with high bioavailability of iron tend to displace lead from the carrier, preventing its absorption. Therefore, in order to reduce the risk of lead poisoning, dietary supplementation with iron has been proposed [43,44].

However, in our sample, the group with iron deficiency (ID) presented BLL similar to the non-iron deficiency group, which contrasts with prior evidence of greater lead toxicity among persons with ID [9,43,44]. However, our findings are endorsed by an earlier study [45] which also found no relationship between toxic BLL and ID. These authors explained the absence of association as being due to the fact that the degree of ID observed was mild in all cases and therefore insufficient to induce a state of lead toxicity.

In our study, the effect of BLL on the haematological and biochemical parameters of iron was also evaluated. The definition of ID was adapted to incorporate new parameters of proven diagnostic utility, namely, CHr, sTfR and the sTfR-F Index [16]. The mean values for these parameters revealed no significant differences between the groups with toxic and non-toxic BLL (Table 3), probably because the maximum BLL obtained was 15 µg/dL, which was insufficient to produce changes in the iron parameters. However, levels of erythrocyte protoporphyrin were significantly higher in subjects with toxic BLL. In this respect, lead is known to interfere with the synthesis of the heme group, preventing the insertion of iron in the protoporphyrin ring by affecting the ferrochelase, thus provoking an increase in free protoporphyrin [46].

Evaluation of erythropoietic activity (reticulocytes, sTfR, EPO) revealed no significant differences between the toxic and non-toxic BLL groups. In adults, decreased erythropoiesis has been reported in individuals with chronic lead poisoning in relation to renal toxicity resulting in decreased EPO production [47]. However, these findings have not been observed elsewhere [1]. In our case, although EPO values were lower in the group with toxic BLL, the differences were not significant, and we located no similar studies in children with which to compare our results.

In the present study, the following limitations are acknowledged: (a) in using graphite furnace atomic absorption spectrometry to determine BLL, and expressing the results of this analysis in absolute numeric values with a lower limit of 1 µ/dL, the mean values obtained might have been over-estimated; (b) a more comprehensive study might have been conducted, taking other risk factors into consideration; (c) although the anamnesis and physical examination performed did not reveal findings suggestive of lead toxicity, the questionnaire design was not aimed at detecting mild symptoms typical of subclinical toxicity.

However, the study made several valuable contributions to our understanding of the questions considered: (a) it was conducted by reference to a large, representative paediatric population, with a wide age range (1–16 years); (b) the study population reflected a broad ethnic and social makeup, and therefore the conclusions drawn can be extrapolated to other cities or countries with similar characteristics; (c) novel analytical variables related to iron metabolism (CHr and sTfR) were included and the degree of erythropoietic activity (which is rarely reported in paediatric studies) was evaluated; (d) the multivariate nature of this study helped us avoid possible confounders in the associations considered.

## 5. Conclusions

In conclusion, in the paediatric population considered, in south-eastern Spain, BLL and the prevalence of toxic BLLs were low and similar to those found in other developed countries. In view of the association observed between toxic BLL in the immigrant population, a low cultural level of the parents and by dietary iron deficiency, health authorities should screen the population in risk sectors and establish preventive and control measures to prevent situations of lead toxicity and its adverse effects on children. Toxic BLL did not produce significant changes in the analytical parameters studied.

## Figures and Tables

**Table 1 ijerph-18-01825-t001:** Charasteristics of subjetcs.

VARIABLES	Nº (%)
**Gender**	
Male	735 (51.5)
Female	692 (48.5)
**Age**	
1–3 year.	346 (24.2)
4–11 year.	684 (48.0)
11–16 year	397 (27.8)
**Origin**	
Spanish	1275 (89.3)
Immigrant	152 (10.7)
**Socio-Economic Level**	
Class 1: High-Income	235 (16.5)
Class 2: Middle-Income	718 (50.3)
Class 3: Low-Income	474 (33.2)
**Parental Education Level**	
No Education/Primary	714 (50)
Secondary/Universitary	713 (50)
**Low Iron Bioavailability Diet**	
No	1150 (80.6)
Yes	277 (19.4)
**Iron deficiency**	
No	1304 (91.4)
Yes	123 (8.6)
**Nutritional Status**	
Normal weight	1008 (70.5)
Overweight/Obesity	419 (29.4)

**Table 2 ijerph-18-01825-t002:** BLL comparison and prevalence of toxic BLL according to the variables studied.

VARIABLES	Nº	BLL µ/dL	*p*	BLL ≥ 5 µ/dL	BBL < 5 µ/dL	OR (95%CI)	*p*
		Mean ± SD (95% CI)		*n* (%)	*n* (%)		
**Gender**							
Male	735	2.0 ± 1.46 (1.9–2.1)	0.29	18 (2.6)	627 (97.4)	1.3 (0.63–2.66)	0.49
Female	692	1.94 ± 0.83 (1.9–2.0)		13 (2.0)	679 (98.0)	1	
**Age**							
1–3 year.	346	2.1 ± 0.6 (2.0–2.1)	0.0001	5 (1.4)	341 (98.6)	0.25 (0.09–0.6)	0.005 0.0001
4–11 year.	684	1.59 ± 0.6 (1.5–1.6)		3 (0.5)	681 (99.5)	0.22 (0.02–0.28)	
11–16 year.	397	2.46 ± 1.8 (2.3–2.6)		23 (5.8)	374 (94.2)	1	
**Origin**							
Spanish	1275	1.89 ± 1.1 (1.8–1.9)	0.0001	10 (0.8)	1265 (99.2)	1	0.0001
Immigrant	152	2.70 ± 1.8 (2.4–3.0)		21 (14.7)	131 (85.3)	20.2 (9.3–44)	
**Socio-Economic Level**							
Class 1	235	1.81 ± 0.5 (1.7–1.9)	0.0001	6 (0.6) 25 (5.3)	947 (99.4) * 449 (94.7)	1 8.1 (3.3–20)	0.0001
Class 2	718	1.8 ± 0.6 (1.75–1.84)					
Class 3	474	2.3 ± 1.2 (2.2–2.4)					
**Parental Education Level**							
No Education/Primary	714	2.1 ± 1.0 (2.05–2.2)	0.0001	26 (3.7)	688 (96.3)	8.6 (2.6–28.6)	0.0001
Secondary/Universitary	713	1.8 ± 0.5 (1.7–1.83)		5 (0.7)	708 (99.3)	1		
**Low Iron Bioavailability Diet**							
No	115	2.1 ± 1.0 (1.95–2.2)	0.0001	15 (1.3)	1135 (98.7)	1	0.002
Yes	277	1.9 ± 0.7 (1.8–1.9)		16 (5.8)	261 (94.2)	3.8 (1.5–9.5)	
**Iron Deficiency**							
No	1304	1.97 ± 1.23 (1.8–1.9)	0.32	28 (2.2)	1276 (97.8)	1	0.35
Yes	123	2.1 ± 0.90 (1.8–2.15)		3 (2.8)	120 (97.5)	0.61 (0.22–1.7)	
**Nutritional Status**							
Normal weight	1008	2.0 ± 1.3 (1.93–2.05)	0.09	24 (2.4)	984 (97.6)	0.71 (0.3–1.7)	0.43
Overweight/Obesity	419	1.9 ± 1.0 (1.8–2.0)		7 (1.7)	412 (98.3)	1	

BLL: Blood lead level; SD: standard deviation; CI: confidence interval: *p*: signification level; OR: odds ratio; Class 1: High-Income; Class 2: Middle-Income; Class 3: Low-Income; *: Socio-economic level Class 1 + Class 2.

**Table 3 ijerph-18-01825-t003:** Comparison of analytical variables between toxic and non-toxic BLLs.

VARIABLES	Total Group (*n*: 1427)	BLL < 5 µg/dL (*n*: 1396)	BLL ≥ 5 µg/dL (*n*: 31)	*p*
	Mean ± SD (95%CI)	Mean ± SD (95%CI)	Mean ± SD (95%CI)	
Haemoglobin (g/dL)	13.3 ± 0.99 (13.2–13.3)	13.3 ± 1.0 (13.2–13.3)	13.3 ± 1.1 (12.9–13.7)	0.77
MCV (fL)	77.7 ± 4 (77.5–77.9)	77.7 ± 3.9 (77.5–77.9)	78.9 ± 6.1 (76.6–81.1)	0.30
CHr (pg)	30.8 ± 2.1 (30.7–30.9)	30.8 ± 2.1 (30.6–30.9)	30.7 ± 2.6 (29.8–31.7)	0.94
Reticulocytes (×10^6^/L)	57.4 ± 19.1 (56.4–58.4)	57.4 ± 19.2 (56.4–58.5)	58.7 ± 16.2 (52.7–64.6)	0.72
sEPO (mU/L)	8.9 ± 5.4 (8.6–9.2)	9.0 ± 5.4 (8.7–9.2)	7.6 ± 6.7 (5.1–10)	0.15
sFerritin (ng/mL)	36.7 ± 19.9 (35.6–37.7)	36.6 ± 19.8 (35.5–37.7)	38.7 ± 24 (29.9–47.5)	0.63
EP (µg/dL RBCs)	31.2 ± 20.1 (31.2–33.3)	32.0 ± 20 (30.9–33.1)	44.0 ± 18.3 (37.3–50.7)	0.001
sTrR (mg/L)	1.31 ± 0.38 (1.29–1.33)	1.31 ± 0.38 (1.29–1.33)	1.25 ± 0.46 (1.08–1.42)	0.37
sTfR-F Index	0.92 ± 0.46 (0.89–0.94)	0.92 ± 0.46 (0.89–0.94)	0.91 ± 0.67 (0.66–1.16)	0.94

SD: Standard deviation; 95% CI: 95% confidence interval; *p*: signification level; BBL: blood lead level; MCV: Mean corpuscular volume; sEPO: Serum erythropoietin; CHr: Reticulocyte haemoglobin content; EP: erythrocyte protoporphyrin; sTfR: Serum transferrin receptor; sTfR-F Index: sTfR/log Ferritin.

**Table 4 ijerph-18-01825-t004:** Multivariate Logistic Regression: Dependent variable: toxic BLL.

Variables	B	SE	*p*	OR (95%CI)
Constant	−4.78			
Origin (Inmigrant)	2.4	5.9	0.0001	11.9 (4.5–31.3)
Parental Education Level	1.53	0.64	0.018	4.6 (1.3–16.3)
Low Iron Bioavailability Diet	1.17	1.6	0.018	3.2 (1.2–8.5)

BLL: Blood lead level; B: beta coefficient; SE: Standard error; *p*: signification level; OR: odds ratio; 95%CI: 95% confidence interval.

## Data Availability

The data presented in this study are available in this article.

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
