# Peer review of "Blood Lead Level in a Paediatric Population of South-Eastern Spain and Associated Risk Factors"

_ijerph, 2021, doi:10.3390/ijerph18041825_

Round 1

Reviewer 1 Report

Since the late 1970s, as lead in the environment has decreased, so has the concentration of lead in pediatric blood.  However, It has been reported that low levels of lead exposure cause decreased intelligence and behavioral problems in children.  In addition, lead is recognized to have health effects not only during fetal exposure, but also current exposure. 

This study targets children of a wide range of ages, and it can be evaluated that the contribution is high because it considers current exposure too.  Although authors are investigating a wide range of ages, it is a bit disappointing that this study does not investigate the development of neuropsychiatry.  However, I think it is valuable data to focus on the interaction between lead and iron, investigate factors related to iron metabolism, and report the relationship with BLL. 

It has also been reported that low-level parental education is one of the causes of high BLL.  The results of these studies may improve education for children worldwide, especially in lead-contaminated areas, resulting in reduced lead-induced health hazards.

Author Response

Our study was conducted in parallel to a nutritional study in the paediatric population. For this reason, we have not investigated the neurological effects it causes. But this is my next project. 
I believe that health programmes should be put in place in developed countries in relation to this issue. Europe in the last few years has experienced a large population movement from African and South American regions. Spain in particular. 
Currently lead is not a problem in our environment, but all children born and raised their first years of life in countries with high rates of lead in air and soil, as well as in products, such as batteries or paints, may contain high values in blood. Even if the numbers are small, there is a proven effect on children's health. It is therefore important that health authorities take measures in this population.

Thanks for yor note.

Reviewer 2 Report

This is a solid manuscript describing risk factors for blood lead in a pediatric population. The comments below should help the authors improve the manuscript.

  1. I would retitle the manuscript to: Risk factors for blood lead in a paediatric population…
  2. Abstract: you need to define the 2 groups for the logistic regression when you mention a logistic regression.
  3. Line 2. “loss of relevant results” is vague and confusing. Maybe you mean “missing data”?
  4. Line 6. Table shows 10.7% immigrants, not 15.2% as reported in the text.
  5. It is not clear to me why you do not also include a linear regression. As you argue, we don’t know that 5 is good cut-off for risk. Why not evaluate the association with continuous lead data? It will increase your power and may provide greater insights on the risk factors.
  6. Discussion: you write “can increase the risk of toxic BLL by 11”. 11 what? 11%? 11-fold? This is not clear as written.
  7. Discussion: your paragraph that starts “However, in our study population,…”. Can you support the final sentence? Do you believe this to be true in your study population?
  8. Last line. Check spelling of “possibe”.

Author Response

Changes made to the text are highlighted in blue

I would change the title of the manuscript to: Blood lead risk factors in a paediatric population...: the study although we do define socio-demographic risk factors, does not specifically study other risk factors for lead poisoning: some factor as such as age of the household, parental smoking, frequent consumption of imported products, natural remedies... so we did not feel it was correct to define the study as determining lead risk factors. 
Summary: 2 groups have to be defined for logistic regression when a logistic regression is mentioned: done
Line 2. "loss of relevant results" is vague and confusing. Perhaps you mean "missing data"? done
Line 6. The table shows 10.7% immigrants, not 15.2% as stated in the text.: done
It is not clear to me why it does not also include a linear regression. As you argue, we don't know if 5 is a good cut-off point for risk. Why not assess the association with continuous lead data? It will increase its power and may provide further insights into risk factors.: I`ll speak with my partner of this point.
Discussion: you write "may increase the risk of toxic BLL by 11". 11 what? 11%? 11 times? 11 times? This is not clear as written.; done
Discussion: your paragraph that begins "However, in our study population,...". Can you support the final sentence - do you think this is true in your study population?:It would be better to say in our sample, wouldn't it?
Last line. Check the spelling of " possibe": done

Reviewer 3 Report

There is a well-done analysis of the observations on the effect of lead and its content.
The discussion is comprehensive and deep, showing the authors' great erudition in this field. The conclusions were correctly drawn from the results. References are up to date and properly selected. However, I would also suggest interesting works by other authors:
1. general lead problem:
- https: // doi: 10.3390 / ijerph17124385
- doi: 10.1515 / intox-2015-0009.
- doi: 10.1016 / j.envres.2016.03.007
- doi: 10.1002 / jcb.26234

2. Or any pathological disorders during pregnancy or even miscarriages https: // doi: 10.7754 / Clin.Lab.2017.170611

Overall, I have no major comments or reservations about this precious paper again.

Minor Notes:
- Please, describe what actions could be proposed by the state health policy and prevention programs in a given field. Are there any implemented?
- tables 2 and 3 should include the reference value range for these groups of people (normal value?)
- I suggest separating Overweight and Obesity to give more precise BMI values
- the listed foods that have been eaten are missing
- and whether there was additional supplementation

Author Response

  • Thank you for your contributions. I will reply to your notes below. The changes made are highlighted in blue in the text:
    • Please, describe what actions could be proposed by the state health policy and prevention programs in a given field. Are there any implemented? done
      - tables 2 and 3 should include the reference value range for these groups of people (normal value?) We found it very complicated to include all the normal values of the parameters in the table, as it is a large table. The normal values are those established by Vazquez et al. in their study. 
      - I suggest separating Overweight and Obesity to give more precise BMI values: As there is no significant association, we do not believe it is relevant to separate by BMI ranges.
      - the listed foods that have been eaten are missing: done
      - and whether there was additional supplementation: Patients on iron treatment were an exclusion criterion. The study subjects were not found to be taking any other type of supplement.
